# Nanoformulations for the Delivery of Dietary Anthocyanins for the Prevention and Treatment of Diabetes Mellitus and Its Complications

**DOI:** 10.3390/ph16050736

**Published:** 2023-05-12

**Authors:** Ana R. Nunes, Elisabete C. Costa, Gilberto Alves, Luís R. Silva

**Affiliations:** 1CICS-UBI, Health Sciences Research Centre, University of Beira Interior, Av. Infante D. Henrique, 6200-506 Covilhã, Portugal; araqueln@gmail.com (A.R.N.); elisabeterochacosta@gmail.com (E.C.C.); gilberto@fcsaude.ubi.pt (G.A.); 2CNC—Centre for Neuroscience and Cell Biology, Faculty of Medicine, University of Coimbra, 3004-504 Coimbra, Portugal; 3CPIRN-UDI-IPG—Research Unit for Inland Development, Center for Potential and Innovation of Natural Resources, Polytechnic of Guarda, 6300-554 Guarda, Portugal; 4CIEPQPF, Department of Chemical Engineering, University of Coimbra, Pólo II—Pinhal de Marrocos, 3030-790 Coimbra, Portugal

**Keywords:** diabetes mellitus, diet, bioactive compounds, anthocyanins, nanoformulations

## Abstract

Diabetes mellitus (DM) is a metabolic disease characterized by abnormal blood glucose levels-hyperglycemia, caused by a lack of insulin secretion, impaired insulin action, or a combination of both. The incidence of DM is increasing, resulting in billions of dollars in annual healthcare costs worldwide. Current therapeutics aim to control hyperglycemia and reduce blood glucose levels to normal. However, most modern drugs have numerous side effects, some of which cause severe kidney and liver problems. On the other hand, natural compounds rich in anthocyanidins (cyanidin, delphinidin, malvidin, pelargonidin, peonidin, and petunidin) have also been used for the prevention and treatment of DM. However, lack of standardization, poor stability, unpleasant taste, and decreased absorption leading to low bioavailability have hindered the application of anthocyanins as therapeutics. Therefore, nanotechnology has been used for more successful delivery of these bioactive compounds. This review summarizes the potential of anthocyanins for the prevention and treatment of DM and its complications, as well as the strategies and advances in the delivery of anthocyanins using nanoformulations.

## 1. Introduction

Diabetes mellitus (DM) is a metabolic disease caused by deficient insulin secretion or impaired insulin action, resulting in abnormal blood glucose levels [1]. Lack or inadequate treatment of this disease can lead to various complications such as cardiovascular disease, coronary heart disease, hypertension, obesity, neurological disorders, diabetic retinopathy and nephropathy, atherosclerosis, hyperlipidemia, and also skin complications [2].

DM is a global burden due to its high morbidity and mortality and imposes high public health costs. According to Kumar and colleagues [3], approximately 463 million people suffered from DM, in 2019. However, these numbers may be even more alarming, as it is estimated that there will be 578 million diabetics worldwide by 2030, and as many as 700 million by 2045 [3]. These statistics reflect the sedentary lifestyle and poor dietary habits in modern societies.

Type 1 diabetes mellitus (T1DM) and type 2 diabetes mellitus (T2DM) are the most common types of diabetes, with T2DM accounting for 90–95% of DM cases [4]. Treatment of DM is performed with synthetic drugs, which can cause serious side effects due to their continuous use and do not provide a cure for the disease. More recently, the global pandemic caused by severe acute respiratory syndrome coronavirus 2 (SARS-CoV-2), commonly known as COVID-19, also affected negatively infected diabetic patients [5]. These patients were less sensitive to the effects of antidiabetic drugs and were also more susceptible to adverse effects [6]. Therefore, there is an increasing need to find and develop new therapeutic agents that are non-toxic, more effective, safer, and less expensive for the treatment of DM. In this context, plants and fruits are emerging as a source of bioactive compounds with biological properties, including antioxidant and antidiabetic activity [7,8,9,10].

The knowledge of traditional medicine has increased the importance of searching for plants and their derivatives with biological health-promoting properties [3,11]. According to World Health Organization (WHO), there are more than 400 plants with antidiabetic properties, but only some of them have been medically and scientifically proven [3]. Phenolic compounds are secondary metabolites of plants and represent the largest class of phytochemicals in edible plants, fruits, and vegetables [12]. Several studies have already shown that these compounds possess interesting biological activities, that are excellent for the treatment of various diseases, including DM [9,13]. Among the various phenolic compounds, anthocyanins have been studied by the scientific community [14,15,16]. These compounds belong to the flavonoids subclass and are glycosides of anthocyanidins, being the pigments responsible for the deep red, purple, and blue colors of many fruits and vegetables [16]. Cyanidin, delphinidin, malvidin, pelargonidin, peonidin, and petunidin are the most important anthocyanidins found in human nutrition [15]. Due to their special properties and various biological activities, these phenolics are used in the food processing, clean energy, and pharmaceutical industries [17]. Numerous studies have reported the numerous biological activities of anthocyanins, such as anti-inflammatory, antioxidant, and anticancer properties [14,18,19,20]. Regarding antidiabetic potential, it has been described that anthocyanidins and their glycosides can inhibit the activity of α-amylase and α-glucosidase, and increase the expression of glucose transporter 4 membrane (GLUT 4), thus promoting glucose uptake and improving lipid profile [16,18]. However, the therapeutic use of anthocyanins presents some challenges, as only a very small proportion (<2%) of the originally ingested anthocyanins are recovered in the bloodstream [21]. This is due to the poor stability and poor bioavailability of these compounds in the gastrointestinal tract [22]. However, nanotechnology can be used to solve these problems. Several studies have shown that the encapsulation of anthocyanins in nanoformulations contributes significantly to the stabilization and bioavailability of this type of phytochemicals [23,24,25,26,27,28].

Although the use of anthocyanin-loaded nanoformulations has been studied mainly for the prevention and treatment of cancer [15,29,30,31], some studies investigated the use of these nanoparticles for the prevention and treatment of DM and its complications. Therefore, this review aimed to provide a general overview of (i) anthocyanins and their antidiabetic potential, and (ii) the application of nanotechnology in the prevention, diagnosis, and treatment of DM, focusing on the studies that described the use of nanoformulations to delivery anthocyanins.

## 2. Diabetic Status: Impact on World Health

DM is a chronic metabolic disorder characterized by hyperglycemia due to deficiency of insulin secretion or impaired insulin action, affecting millions of people worldwide. In the next few years, diabetes prevalence is expected to increase by 20% in developed countries, while in developing countries it is expected to increase by 69% [32]. Globally, more than 400 million people have diabetes, and this number is expected to increase to 578 million by 2030 [3].

The American Diabetes Association (ADA) defines diabetes as fasting plasma glucose ≥ 126 mg/dL, a 75 g oral glucose tolerance test, 2-h plasma glucose > 200 mg/dL, or hemoglobin A1c ≥ 6.5% [33]. DM is mainly divided into T1DM and T2DM, with T2DM being the most common. T1DM is insulin-dependent diabetes and occurs when pancreatic *β*-cells are damaged, resulting in deficient production of this hormone. In addition, this type of diabetes can be classified as idiopathic or immune-mediated [34]. In T1DM, there may be irregular hyperglycemia, a risk of severe hypoglycemia, and diabetic ketoacidosis. T2DM accounts for 90% to 95% of cases and is caused by insulin resistance in the liver and other peripheral tissues [34]. Furthermore, these people’s ongoing hyperglycemia metabolic changes, inflammation, and cellular death. In addition, to other cardiovascular diseases, nephropathy, dyslipidemia, and oxidative stress are frequently linked to the mortality and morbidity rates of T2DM [35]. WHO reports that the age-specific death rates for diabetes increased by 3% between 2000 and 2019 [36]. It is concerning that mortality rates are rising globally and that incidence rates are rising as well. Additionally, people with diabetes have greater mortality rates from sepsis and a higher chance of developing infections, such as pneumonia, otitis, and urinary tract infections [35].

DM is currently in its prodromal stage, also referred to as the prediabetic state. Prediabetes, which has blood glucose levels that are higher than normal but lower than those of diabetic patients, is reversible and less severe than DM [37]. The people in this state are also insulin resistant and/or glucose tolerant [38]. Approximately 50% of those with prediabetes will develop T2DM within seven years, and 83% will convert throughout the course of their lifetime, according to data analyses [39]. An increase in oxidative stress and, as a result, a decrease in antioxidant defenses are both linked to DM and prediabetic conditions [37,38].

Over the past few years, there has been a continuous hunt for innovative DM prevention and treatment methods. One or more of the solutions is the use of natural products, a healthy diet, regular exercise, and blood pressure and lipid profile regulation [40]. In this context, medicinal plants are abundant in many bioactive compounds, including phenolics like anthocyanins, which have already demonstrated promising antioxidant and antidiabetic effects [7,8,10,13].

## 3. Anthocyanins: A Potential Natural Antidiabetic

The main sources of phenolic compounds include plants, fruits, and vegetables. The primary classes of phenolics are flavonoids and non-flavonoids, and they have already been demonstrated to have bioactive effects against a variety of diseases [41]. These phytochemicals are promising agents to be used in the pharmaceutical, cosmetic, and food industries due to the variety of their chemical structures [10,13]. Flavonoids have been emphasized as phenolic substances with significant biological activities [7,8,42,43]. They are low-molecular-mass phenolic secondary compounds of plants that aid in defending plants from environmental stresses [44]. Additionally, flavonoids are recognized as floral pigments [45]. These compounds are made of two aromatic rings (A and B) and a heterocyclic ring (C) with an oxygen atom, each with a 15-carbon skeleton (C_6_-C_3_-C_6_) [14].

Phenolic substances are well known for having properties that support health, such as antioxidant, antidiabetic, antimicrobial, anticancer, and others [9,13,46,47]. Anthocyanins have been shown to play a significant part in the process of metabolic diseases such as DM [39], with epidemiological research demonstrating an inverse relationship between dietary flavonoids and T2DM incidence [48,49,50].

### 3.1. Structure and Function

The family of flavonoids known as anthocyanidins and their glucosides, also known as anthocyanins (*anthos* means flower and *kyanos* means blue), produced via the phenylpropanoid pathway, are what give many fruits, vegetables, and beverages their deep red, purple, and blue colors [16]. They contribute to the nutritional and sensory properties of plants and are water-soluble [14]. These compounds may also function as pollinators, antifeedants, and phytoalexins, in addition to aiding in a plant’s defense against pathogens, predators, UV radiation, and environmental factors [14]. Blueberries, cherries, raspberries, strawberries, purple grapes, black currants, and red wine are the main sources of anthocyanidins [16].

Chemically speaking, anthocyanins are found as a glycoside that contains both a non-sugar component (aglycone: anthocyanidin) and sugar (glycone moiety: glucose, galactose, xylose, rhamnose, or arabinose) [15]. These substances are flavonoids because they have three benzoic rings: an A ring, a heterocyclic ring with an oxygen atom (C ring), and a B ring that is benzoic with a carbon-carbon bond link called flavylium ion [51]—Figure 1.

The most prevalent anthocyanidins discovered in food are cyanidin, delphinidin, malvidin, pelargonidin, peonidin, and petunidin—Table 1, which are among the approximately 30 anthocyanidins currently known [14,19]. The classification of anthocyanins is made according to (i) the number, position, and degree of methylation of the hydroxyl groups; (ii) the number and nature of the sugar moieties bonded to the aglycone; and (iii) the position of the aliphatic and/or aromatic carboxylate acids on the sugar molecule [14]. Some of the elements accountable for the various biological activities of these compounds include hydroxylation, methylation, and the number and type of sugars linked to the aglycone [14,15,52]. Anthocyanins are powerful antioxidants because of their chemical structure. The anthocyanin skeleton’s hydroxyl (-OH) and methoxy (-OCH_3_) group count and location both have an impact on the antioxidant potential of the substance. For instance, the antioxidant activity is greater when there are more hydroxyl groups [14]. Additionally, cyanidin, delphinidin, and pelargonidin are highly effective against the superoxide anion, while pelargonidin is effective against hydroxyl radicals [53].

Anthocyanins are less stable due to factors such as temperature variations, cooking, exposure to light and oxygen, as well as the presence of enzymes, phenolic compounds, metal ions, ascorbic acid, hydrogen peroxide, and water [54]. They are also influenced by storage and processing conditions.

### 3.2. Main Sources

Anthocyanins can be found in large amounts in many red and blue fruits and vegetables—Table 1. Their content depends on the species, cultivar, growing region, climate (e.g., temperature, humidity, light exposure), harvesting, ripening, processing, and storage conditions [16]. The main sources of anthocyanins are berries, like strawberries, blueberries, blackberries, blackcurrant, and raspberries [16]. These berries contain between 100 and 700 mg of anthocyanins per g of fresh fruit [55,56]. Elderberries and chokeberries have the greatest concentrations of these compounds (1.4 to 1.8 g per 100 g). The skin of cherries has the highest concentration of anthocyanins, followed by flesh and pits [57]. Other great sources include açai, purple corn, plums, pomegranates, eggplant, wine, grapes, and red/purple vegetables [55,56,58].

The Mediterranean diet, which is high in these phenolic compounds due to its abundance in red fruits and wine, is the one with the greatest daily intake of anthocyanins [59]. About 70% of the anthocyanins consumed every day come from fruits, while 25% come from wine 25% [60]. The main anthocyanidins consumed by humans are cyanidin, delphinidin, malvidin, pelargonidin, peonidin, and petunidin, with cyanidin 3-*O*-glucoside being primarily found in berries, and malvidin 3-*O*-glucoside in broad varieties of foods [16]. To guarantee an adequate level of bioactive compounds with health-promoting properties, regular consumption of fruits and vegetables is imperative. Numerous studies have shown that eating foods high in phenolic compounds, such as anthocyanins, can reduce oxidative stress and inflammation, which lowers the chance of developing chronic diseases [13,14,47,59].

### 3.3. Antidiabetic Potential

Anthocyanins have been found to be beneficial in the prevention and treatment of DM and its complications, according to several investigations [9,17,61]—Table 2. These compounds have demonstrated the ability to reduce hyperglycemia, insulin resistance, reactive species, and proinflammatory cytokines in this setting [8,9,17,48]. Additionally, they were found to be involved in gluconeogenesis suppression, as well as α-amylase and α-glucosidase activity [62,63,64].

The enzymes α-amylase and α-glucosidase hydrolyze carbohydrates and produce glucose, and thus they are crucial for controlling digestion and absorbing glucose. Numerous studies have demonstrated the ability of anthocyanins or the consumption of foods high in anthocyanins to inhibit these enzymes, thereby modulating postprandial blood glucose and preventing the onset of DM [8,62,64,65]. The α-glucosidase enzyme can be inhibited by sweet cherry extracts, according to previous research [8]. The authors claim that cherries are extremely rich in anthocyanins, especially cyanidin 3-*O*-rutinoside [8], which has already been shown to significantly inhibit α-glucosidase activity in a concentration-dependent way [66]. According to additional research, cyanidin from *Cinnamomum camphora* fruit inhibited α-glucosidase action more potently than Acarbose (a drug used for the management of glycemic control in patients with T2DM) [67]. The ability to inhibit α-glucosidase was also demonstrated in other experiments with blueberry, blackcurrant, and blue honeysuckle [68], blackberry [69], and bilberry and cranberry [70]. The activity of pancreatic α-amylase was shown to be inhibited by cyanidin-3-rutinoside in research conducted by Akkarachiyasit and colleagues [71]. According to recent research on animal studies, T2DM mice fed blackcurrant extract, which contains high levels of delphinidin 3-rutinoside, demonstrated a reduction in blood glucose concentration and an improvement in glucose tolerance [72]. Another study found that cyanidin 3-glucoside reduced fasting blood glucose and increased glycogen synthesis, which was most likely caused by an increase in GLUT-1 expression in the liver of *db/db* mice [73]. Malvidin, malvidin 3-glucoside, and malvidin 3-galactoside, which are all components of blueberry anthocyanin extract, were discovered by Herrera-Balandrano and co-works [74] to be capable of inhibiting diabetes hyperlipidemia and decrease insulin levels. In streptozotocin-induced diabetic rats treated for 12 weeks, anthocyanins from purple sweet potatoes improved blood glucose and lipid levels and reduced oxidative stress and liver damage [75].

In human studies, anthocyanins’ ability to treat diabetes was also assessed. A randomized controlled study performed on 37 people with T2DM demonstrated that taking 350 mg of whortleberry fruit hydroalcoholic extract, every eight hours for two months could lower blood levels of fasting glucose, 2-h postprandial glucose, and HbA1C [76]. Similarly, eating freeze-dried strawberries for six weeks increased antioxidant capacity and blood glucose levels, while lowering inflammatory reaction and lipid peroxidation [77]. Anthocyanin supplementation for 12 weeks improved serum adiponectin and fasting glucose in patients with recently diagnosed diabetes, according to a study that included people with prediabetes or diabetes [78].

Despite a number of scientific studies showing that anthocyanins are crucial for the prevention and treatment of DM and its complications—Table 2, it is important to take into account that the limited bioavailability and poor stability of these colored compounds hinder the achievement of their highest therapeutic potential. To improve the efficacy of anthocyanins, lipids, polysaccharides, and protein complexes or nanoencapsulation may be suitable substitutes.

**Table 2 pharmaceuticals-16-00736-t002:** In vitro studies, animal studies, and clinical trials on the antidiabetic potential of anthocyanins.

Source	Anthocyanin Type	Main Outcomes	Reference
Sweet Cherries (*Prunus avium* L.)	Anthocyanins-enriched fraction	α-glucosidase inhibition	[8]
*Cinnamomum camphora* L. fruit	Cyanidin	α-glucosidase inhibition	[67]
Blueberry, blackcurrant and blue honeysuckle fruits	Anthocyanins-enriched fraction	α-glucosidase inhibition	[68]
Blueberries (*Vaccinium corymbosum*) and blackberries (*Rubus* spp.)	Anthocyanins-enriched fraction	α-glucosidase inhibition	[69]
*Vaccinium oxycoccos* L. and *Vaccinium myrtillus* L.	Anthocyanins-enriched fraction	α-glucosidase inhibition	[70]
n.d.	Cyanidin 3-rutinoside	α-amylase inhibition ↓ postprandial glycemia	[71]
Blackcurrant extract (11 g per kg)	Anthocyanins-enriched fraction	↓ blood glucose ↑ glucose tolerance	[72]
n.d.	Cyanidin 3-*O*-glucoside	↓ fasting blood glucose levels ↓ accumulation of liver lipids ↑ glycogen synthesis	[73]
Blueberry anthocyanin extract (100.4 mg per kg)	Anthocyanins-enriched fraction	↓ fasting blood glucose levels ↓ insulin levels ↑ liver antioxidants	[74]
Purple sweet potato	Anthocyanins-enriched fraction	↓ blood glucose levels ↑ glucose tolerance ↓ liver damage ↑ antioxidant capacity	[75]
Whortleberry fruit hydroalcoholic extract (1.0 g per day)	Anthocyanins-enriched fraction	↓ blood glucose of fasting glucose	[76]
Freeze-dried strawberry (100 g per day)	Anthocyanins-enriched fraction	↓ lipid peroxidation ↓ HbA1c and total antioxidant status	[77]
n.d.	Anthocyanins (320 mg per day)	↑ adiponectin ↓ fasting glucose ↓ basal glycemia and insulinemia	[78]

## 4. Application of Nanotechnology in DM

Nanotechnology includes the atomic- or molecule-level manipulation of materials in physics, chemistry, and biology [79]. Nanotechnology has so far greatly benefited our society, making great contributions in several prestigious industries, including electronics, energy, agriculture, textile, environmental remediation, cosmetics, medicine, and healthcare, among others [80]. The advancement of nanotechnology in the medical and healthcare fields over the past 20 years has helped with the prevention, diagnosis, and treatment of numerous pathologies, including DM [79,81].

Up to now, nanotechnology has been widely investigated for the prevention, diagnosis, and treatment of DM and its complications, as previously reviewed in detail in several articles [82,83,84,85,86,87,88,89,90,91,92]. Nanotechnology has aided in the diagnosis of DM, as the early detection of this disease or knowledge of its exact stage of progression is essential for its management and for the prevention of its potentially fatal complications. For instance, since the loss in β-cell mass is what drives the progression of DM, specifically type 1, nanoprobes (e.g., metallic and magnetic nanoparticles) with β-cell specificity and high contrast can quantify β-cell mass by using various imaging techniques, such as computed tomography (CT), positron emission tomography (PET) and magnetic resonance imaging (MRI), which is otherwise only possible by performing a post-mortem autopsy [84]. Moreover, new nanotechnology-based sensors, such as metal nanoparticles optical sensor systems, carbon nanotubes for fluorescent glucose sensing, and nanoparticles for direct enzyme-free detection of glucose, among others, enable the accurate gathering of data about the patient’s blood glucose levels and, as a result, better insulin dosing [92]. Additionally, “artificial pancreas” or “‘smart” glucose-responsive insulin-based therapies have been developed using nanotechnology. Examples of these devices include patches that contain microneedles packed with insulin and glucose-sensing enzymes that are released in response to the body’s needs, thereby lowering the frequency of hypoglycemic and hyperglycemic events [84,93]. Additionally, this eliminates the need for patients to conduct routine examinations, which can be challenging and uncomfortable, especially for young children and the elderly.

Last but not least, using nanoformulations to deliver antidiabetic drugs can greatly aid in the prevention and treatment of DM [82,85,94]. Examples of such formulations include ceramic nanoparticles, liposomes, dendrimers, polymeric biodegradable nanoparticles, and polymeric micelles. In short, nanoformulations effectively transport the antidiabetic agent to the target site in the desired release pattern while also enhancing its stability. Utilizing nanocarriers can also lower medication dosage and administration frequency, likely reducing the risk of toxic effects.

### 4.1. Nanoformulations for the Delivery of Anthocyanins for the Prevention and Treatment of DM

Numerous in vitro and in vivo studies have shown that bioactive compounds found in nature, such as anthocyanins, aid in the prevention and treatment of metabolic disorders like DM, as previously discussed in this review—Table 2. In light of this, it is advised to consume anthocyanins supplements or anthocyanin-rich foods (berries, currants, grapes, tropical fruits, etc.) as a preventive and therapeutic approach towards DM, and even DM-related complications, like nephropathy, neuropathy retinopathy, hearts problems, and among others [57,95,96]. Because the traditional antidiabetic medications (Insulins, Sulfonylureas (SUs), Thiazolidinediones (TZDs), Biguanides, Meglitinides, etc.) are linked to a number of side effects, including gastrointestinal disorders such as constipation, diarrhea, vomiting, nausea, and even severe hepatic or renal impairment, the use of these naturally-derived products has also recently attracted significant attention [97]. However, a number of variables limit the potential of natural-derived compounds. In the case of anthocyanins, their low in vivo absorption, which is brought on by their poor stability and poor bioavailability in the gastrointestinal tract, hinders their therapeutic impact [22]. Anthocyanins are vulnerable to gastrointestinal pH, enzymatic environments, and microbiota after ingestion, which cause the anthocyanins to be broken down into metabolites and have poor bioavailability [22,98,99]—Figure 2.

Additionally, the strong polarity of anthocyanins prevents them from crossing the lipid-rich outer membrane of small intestine enterocytes [98]. Even after absorption, complete anthocyanins may continue to be metabolized in the liver or kidney. Thus, by increasing the stability and bioavailability of anthocyanins in the gastrointestinal tract, nanoformulations can be used to greatly increase anthocyanins’ absorption (Figure 3).

According to a recent review by Shen et al. [98], nanoformulations (lipid-based, polysaccharide-based, and protein-based complexes, as well as in anthocyanin-encapsulation systems) have been researched to deliver anthocyanins and can lessen anthocyanins’ high instability and poor bioavailability in a variety of ways.

Anthocyanins are compounds with high polarity, which limits their crossing through the lipid-rich outer membrane of the small intestine. Having this in mind, lipid-based nanoformulations (liposome, noisome, micelle, etc.) with the right balance of lipophilicity and hydrophilicity can increase intestinal transport and thereby increase the bioavailability of anthocyanins [98]. In fact, studies demonstrated that phospholipid-anthocyanins nanoformulations allowed a better transepithelial transport activity than free anthocyanins, prolong the half-life of anthocyanins in the gastrointestinal tract, and increased concentration in blood [100]. Similarly, it was found that noisome allowed a better absorption of anthocyanins inside the intestine *villus* [101]. Nevertheless, phospholipids can easily oxidize [100], and the assembly of niosomes (ratios of cholesterol and surfactant) limits the trapping efficacy and controlled release of anthocyanins [101].

On the other hand, by preventing them from degrading in biological environments, anthocyanins’ conjugation with polysaccharides (such as chitin, cellulose, chitosan, pectin, etc.) can enhance their stabilization. It has been reported that polysaccharides can stabilize anthocyanin pigment through modification of the equilibrium and rate constants of the flavylium network [102,103]. Polysaccharides are additionally renowned for their capacity to regulate the release of substances [104]. For instance, pectin interacts slowly with bile salts [105], and this interaction was stronger than that between pectin and anthocyanins, thus, causing the controlled release of anthocyanins from the pectin complex in the intestine [106]. As a result, they permit the release of anthocyanins for extended durations, increasing their bioavailability. Nevertheless, it must be considered that it was also found some issues related to the polysaccharides-based delivery of anthocyanins. For instance, some polysaccharides, such as chitosan, only dissolve in acidic environments, which may affect the release of anthocyanins [23]. Pectin-cyanidin-3-glucoside nanoformulation also demonstrated low loading efficiency [107].

Anthocyanins can naturally associate with proteins (such as those found in soy, whey, etc.) to create more stable protein-anthocyanin complexes. The anthocyanins-protein interaction is based on the formation of multiple weak interactions, reinforcing and stabilizing the complexes between amino acid side chains of protein and aromatic rings of anthocyanins [98]. Soy protein-blueberry anthocyanin nanoformulations showed better bioaccessibility in simulated jejunal and ileal compartments [108] and whey protein-bilberry anthocyanin capsules increased systemic concentrations and short-term bioavailability of anthocyanins [109]. On the other hand, anthocyanin released in the ileal efflux might be metabolized in the colon instead of entering body circulation [108] and the isoelectric points of whey protein limited their applications [109].

Lastly, anthocyanins can also be encapsulated being therefore protected with various coating materials, usually polymeric materials. These materials potentially improve the anthocyanin’s stability by serving as a barrier towards different factors such as oxygen, light, temperature, water, enzyme, and reactive compounds [110]. Additionally, anthocyanin-encapsulation systems can be produced by controlling certain parameters can be controlled such as place and time of delivery for maximum efficiency, the size of vesicles produced by this method varies [110]. The majority of anthocyanin-loaded nanoformulation research to date has focused on cancer prevention and therapy [15,29,31,44]. However, some studies showed the potential of anthocyanin-loaded nanoparticles for the prevention and treatment of DM, and its complications, as detailed in the following sections.

#### 4.1.1. Anthocyanin-Loaded Nanoparticles for DM Prevention

Because mitochondria are involved in a number of insulin-signaling mechanisms that maintain proper glucose homeostasis, this organelle is crucial in the emergence of DM. Samadder et al. [111] created pelargonidin-loaded poly-lactide-co-glycolic-acid (PLGA) nanoparticles in order to investigate the protective effects of this anthocyanin in preserving glucose homeostasis and mitochondrial functionality in L6 muscle cells under the influence of alloxan (ALX)-induced prediabetic/diabetic states. Results showed that the disruption of enzymatic protein glucokinase, the GLUT4 glucose transporter in ALX-induced L6 cells was associated with an imbalance in glucose homeostasis. This disruption was lessened in cells treated with free pelargonidin and pelargonidin-PLGA nanoparticles before the ALX treatment. Other tests performed in this study, showed that cells that were pre-treated with free-pelargonidin and pelargonidin-PLGA nanoparticles expressed more of the proteins (IRS1, IRS2, and PI3) involved in glucose regulation. Additionally, pelargonidin showed defense against DNA deterioration and reactive stress. Although both free-pelargonidin and pelargonidin-PLGA nanoparticles had beneficial impacts on the development and prevention of DM, pelargonidin-loaded nanoparticles performed more effectively than pelargonidin alone, even at a dose that was 10-fold lower. The authors hypothesized that this finding was caused by pelargonidin’s enhanced intracellular penetration when loaded into nanoparticles, enabling faster transport to target locations.

In diabetic individuals, the enzymes α-amylase and α-glucosidase digest the carbohydrates and raise the postprandial glucose level. Therefore, postprandial hyperglycemia can be controlled and the chance of developing DM can be decreased by inhibiting the activity of these enzymes DM [65]. On the other hand, oxidative stress has a significant impact on T2DM development [112]. Using *Bauhinia variegata* flower extract, which is rich in anthocyanins like cyanidin-3-glucoside, malvidin-3-glucoside, malvidin-3-diglucoside, peonidin-3-glucoside and peonidin-3-diglucoside, Johnson and colleagues [113] showed that silver nanoparticles may prevent diabetes by inhibiting the antioxidant property and α-amylase enzyme activity. For 2,2-diphenyl-1-picrylhydrazyl (DPPH) and ferric-reducing power activity assays, the IC_50_ value of nanoparticle was discovered to be 4.64 and 16.6 µg/mL, respectively. The nanoparticle’s IC_50_ measurement for inhibiting α-amylase was reported to be 38 µg/mL.

#### 4.1.2. Anthocyanin-Loaded Nanoparticles for DM Treatment

The in vitro and in vivo inhibitory effects of delphinidin and cyanidin in the free and liposomal forms on the albumin glycation reaction were examined in a 2013 study by Gharib and colleagues [61]. The inhibition of the protein glycation process is considered a promising strategy to control DM. The authors encapsulated delphinidin in liposomes made of phosphatidylcholines and cholesterol. Results revealed that the albumin glycation was significantly inhibited by anthocyanins in their liposomal form more effectively than in their free form. The rate of albumin glycation with delphinidin and cyanidin chloride in free forms could be reduced to 30.50 and 46.00%, respectively, when compared to control groups. Delphinidin and cyanidin-loaded liposomes were able to lower albumin glycation to 8.50 and 14.60%, respectively, under the same circumstances. The rate of albumin and HbA1c glycation could be lowered to 46.35 and 3.60%, respectively, with daily administration of 100 mg/kg delphinidin-loaded liposomes after 8 weeks of in vivo trials in diabetic mice. Additionally, under the same circumstances, the cyanidin-loaded liposomes could reduce albumin and HbA1c glycation rates to 55.56 and 4.95%, respectively. The writers concluded that these formulations are effective at inhibiting protein glycation and could be used to manage DM [61].

Pelargonidin was also encapsulated into PLGA nanoparticles by emulsion-diffusion-evaporation method in another work by Roy et al. [114]. The effects of free-pelargonidin and nanoparticles were examined in a rat model of diabetes caused by streptozotocin. 0.6 mg pelargonidin/kg body weight or 0.6 mg of pelargonidin-loaded nanoparticles/kg body weight were administered intravenously to rodents at an internal of 3 days. Two doses of nanoparticles were administered, and this helped to manage hyperglycemia and hyperlipidemia, decreased enzymatic antioxidant activity, and elevated oxidative stress markers. Overall, using nanoparticles to deliver pelargonidin enabled better control of the diabetogenic effects in streptozotocin-induced diabetic rats. This is likely because pelargonidin has an improved dissolution rate, slow release, and long-acting impact.

More recently, I’tishom et al. [115] showed that DM can be treated with carboxymethyl chitosan and alginate nanocapsules loaded with anthocyanin-rich purple-sweet-potato-extract. Researchers found that purple sweet potato extract-loaded chitosan and alginate nanocapsules were 4.4 times more effective at lowering blood glucose levels in diabetic mice caused by Streptozotocin than the extract alone.

#### 4.1.3. Anthocyanin-Loaded Nanoparticles for DM Complications

Patients with diabetes mellitus frequently experience hyperlipidemia, which is characterized by abnormally elevated blood levels of fats (lipids), including cholesterol and triglycerides, and is associated with several additional health issues [116]. In order to lessen hyperlipidemic aberrations and consequently DM problems, Sreerekha et al. [117] created anthocyanin-loaded chitosan nanoparticles. In rats fed on high-fat, high-alcohol diets, dietary supplementation with nanoparticles reduced lipid peroxidation, boosted antioxidant enzyme activity and inhibited the growth of lipogenesis. In other words, the group given the nanoparticles had significantly higher catalase and superoxide dismutase (SOD) enzyme activity levels, decreased serum levels of total cholesterol and triglycerides, and decreased lipid-mediated oxidative stress and lipid levels in serum and liver.

Among the many complications of DM, cardiac dysfunction is a serious one [118]. By maintaining glycogen levels in the heart tissue and lowering malondialdehyde and hydroxyproline levels in mice, Hanafy and colleagues [119] were able to effectively attenuated cardiac dysfunction brought on by fibrosis caused by anthocyanin incorporated into hydrogel nanoparticles. This indicates that the delivery of anthocyanin via nanoparticles may increase its retention in the body, because the capsules may protect anthocyanin from immunoglobulins and cellular immune system components.

## 5. Conclusions

Natural bioactive compounds are excellent candidates to be studied due to the ongoing rise in the prevalence of DM around the globe and the need to create new, efficient, safe, and low-toxic medications. Due to their advantageous health properties, anthocyanins have been receiving increasing focus. The chance to use these compounds in the creation of nutraceuticals, functional foods, and cosmetics is made possible by their diversity and the ease with which they can be discovered and acquired in nature. Numerous studies demonstrate that these flavonoids work through a variety of antioxidant and anti-inflammatory pathways to improve insulin resistance, lipid metabolism, glucose metabolism, and insulin sensitivity. To counteract the reduced stability and bioavailability of anthocyanins, numerous attempts have been made. Therefore, using nanoformulations to transport these compounds in the human body may be a good idea. Examples include liposomes, polymeric biodegradable nanoparticles, ceramic nanoparticles, and polymeric micelles. Nevertheless, before the use of nanoparticles in the clinical environment they must check several requirements in terms of their biodegradability, biocompatibility, drug release time, stability, and integrity of biomacromolecules. To better understand and show the potential of anthocyanin-loaded nanoformulations to prevent and treat DM, even more, research must be done, since these nanoformulations have been mostly investigated for cancer prevention and treatment.

## Figures and Tables

**Figure 1 pharmaceuticals-16-00736-f001:**
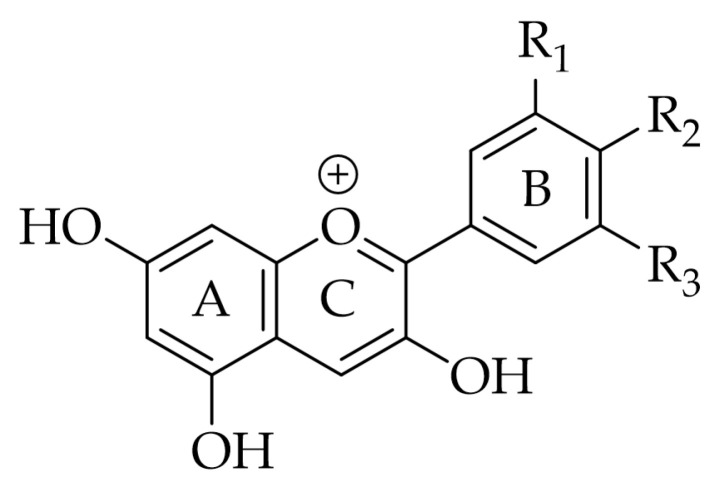
Chemical structure of anthocyanins: two benzoic rings (A) and (B) separated by a heterocyclic (C) ring.

**Figure 2 pharmaceuticals-16-00736-f002:**
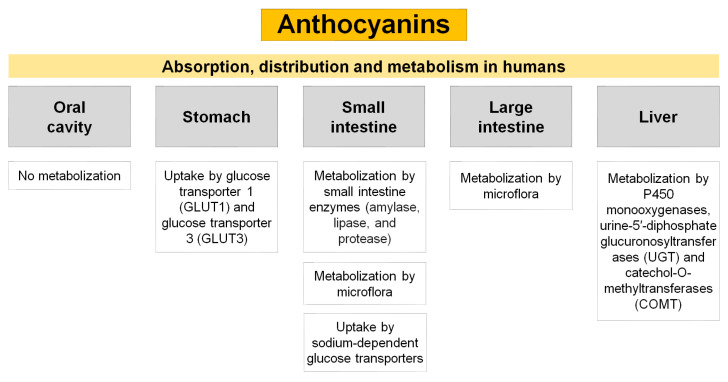
Overview of the anthocyanins’ absorption, distribution, and metabolism in humans.

**Figure 3 pharmaceuticals-16-00736-f003:**
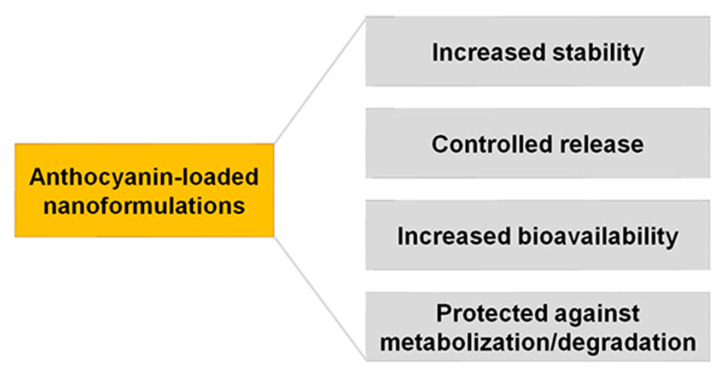
Advantages of the nanoformulations for the delivery of anthocyanins.

**Table 1 pharmaceuticals-16-00736-t001:** Chemical structures and sources of six common anthocyanidins found in nature (adapted from [53]).

Anthocyanidin	R1	R2	R3	Natural Sources
Cyanidin	-OH	-OH	-H	Apple, blackberry, elderberry, plum, peach, nectarine
Delphinidin	-OH	-OH	-OH	Oranges, grapes, beans
Pelargonidin	-H	-OH	-H	Strawberries, red radishes
Malvidin	-OCH_3_	-OH	-OCH_3_	Grapes
Peonidin	-OCH_3_	-OH	-H	Cranberries, blueberries, plums, cherries, grapes, purple corn
Petunidin	-OH	-OH	-OCH_3_	Grapes, red berries

## Data Availability

Data are contained within this article.

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
