# Peer review of "Nanoformulations for the Delivery of Dietary Anthocyanins for the Prevention and Treatment of Diabetes Mellitus and Its Complications"

_pharmaceuticals, 2023, doi:10.3390/ph16050736_

Round 1

Reviewer 1 Report

The authors provides a comprehensive review of the potential of anthocyanins for the prevention and treatment of diabetes mellitus and its complications.

I would like to offer some constructive feedback that I believe may help improve the manuscript:

1. I would suggest a full revision of the manuscript to improve its grammatical quality. Specifically, limiting the use of passive voice and correcting minor typos.

For instance:

In line 362 "enzimes" should be enzymes. Additionally, the whole phrase could be improve for better understanding as follows:

"In individuals with diabetes, the enzymes α-amylase and α-glucosidase digest carbohydrates, leading to an increase in postprandial glucose levels."

Line 307-308 the phrase  "limited bioavailability" is more common instead of "poor bioavailability

2. Authors should consider adding images to their manuscript. Incorporating relevant visuals can not only make the article more visually appealing, but it can also help to clarify and reinforce the key points you are making.

 I would suggest a full revision of the manuscript to improve its grammatical quality. Specifically, limiting the use of passive voice and correcting minor typos.

For instance, in line 362 "enzimes" should be enzymes. Additionally, the whole phrase could be improve for better understanding as follows:

"In individuals with diabetes, the enzymes α-amylase and α-glucosidase digest carbohydrates, leading to an increase in postprandial glucose levels."

I corrected the spelling of "enzimes" to "enzymes," added the missing verb "leading," and rephrased the sentence slightly for clarity.

Author Response

1. The authors are grateful for the reviewer’s comment and suggestions. All the manuscript was revised. (Please see now the revised version of the manuscript).

The phrase in lines 307-308 was revised. (Please see now line 319 of the revised version of the manuscript)

2. Authors agree with the reviewer, i.e., that relevant visuals always contribute for the visual appealing of the manuscript. Unfortunately, authors decided to not add new images to the review. This decision was taken in order to avoid the repetition of information that is already in the text or in other review articles already published.

The authors thank by the reviewer’s comment and suggestions. All the manuscript was revised. (Please see now the revised version of the manuscript)

Reviewer 2 Report

This study provides a concise overview of the potential of anthocyanins for the prevention and treatment of DM and its complications, as well as the tactics and advancements in the administration of anthocyanins using nanoformulations.

The authors should focus more on the common complications of diabetes mellitus and the application of nanotechnology before conducting a fair comparison on Anthocyanin-loaded Nanoparticles for DM Complications.

Include tables that clearly show the type of complication (Diabetic nephropathy, Diabetic retinopathy, Diabetic cardiomyopathy, Peripheral, Diabetic foot, Segmental bone injury, Diabetic macroangiopathy, and so on) and give results from Anthocyanin-loaded Nano systems for these conditions.

Nanotechnology has been tested in a number of illness trials, but there are still a number of issues that need to be resolved before it can be widely used, including biodegradability, biocompatibility, drug release time, stability and integrity of biomacromolecules, and targeting of nanoparticles. Furthermore, the application of nanotechnology in medical care is essential. To better assist clinical illness treatment, additional molecular research are required to verify its safety. So authors should consider all these points and report any benefit or target achieved by anthocyanin loaded nano systems or not?

Please check the fluency of sentences

Author Response

1. 

The authors acknowledge the reviewer for raising this concern about the importance of the overall application of the nanotechnology in DM. Authors described in the first version of the manuscript (section “4. Application of Nanotechnology in DM”) the use of nanotechnology in the prevention, diagnosis and treatment of DM. Authors tried to discuss this topic lightly since several reviews in the literature already described extensively the application of nanotechnology in DM, and therefore we wanted to be more focused on the use of the nanoformulations.

Nevertheless, we cited in the revised version of the manuscript several review articles to further elucidate the reader about the use of nanotechnology in general for the prevention and treatment of DM and its complications. (Please see now lines 268 to 270 of revised manuscript).

2. The authors thank the reviewer’s comment. Authors agree with the reviewer, i.e., there is a lot of investigation and development that must be performed before nanoformulations could be marketed and used widely in a clinical environment. To further elucidate about this issue, authors modified the “5. Conclusions” section (Please see now lines 474 to 479 of revised manuscript).

3. The text was revised in order to improve the English both grammatically and syntactically, but also to eliminate inconsistencies.

Reviewer 3 Report

The manuscript, titled "Nanoformulations for the Delivery of Dietary Anthocyanins for the Prevention and Treatment of Diabetes Mellitus and its Complications," describes the potential clinical application of anthocyanins in the prevention and therapy of diabetes, as well as the use of nanoencapsulation as a major approach to enhance their pharmacological potential and clinical application.

Although the manuscript is generally well written to improve its comprehensiveness, some issues need to be improved.

1. The main advantages of nanocarriers to address the unfavorable bioavailability associated with anthocyanins are not well emphasized.

2. The main groups of nanocarriers with their advantages are also not discussed, which interferes with the logically coherent structure of the manuscript and its focus is shifted to diabetes as a disease rather than the advantages and need for developing nanoformulations of anthocyanins (which is the main focus of the title).

Minor editing of English language required  

Author Response

1. 

Authors would like to refer that, to the best of our knowledge, the advantages/disadvantages of different nanoformulations for the delivery of anthocyanins is very limited/absent. On the other hand, unfortunately, the research articles that investigated the use of anthocyanin-loaded nanoformulations for DM did not perform bioavailability assays. In fact, most of the available information about the delivery of anthocyanins using nanoformulations resulted from cancer related studies, as referred in this review (Please see now lines 328 to 3331 of revised manuscript).

2. Nevertheless, following the reviewer suggestions, authors modified the manuscript in order to better describe as possible: i) the types of nanoformulations used to deliver anthocyanins; ii) the pros and cons of different nanoformulations in the delivery of anthocyanins. Please see the performed modifications in section “4.1. Nanoformulations for the Delivery of Anthocyanins for the Prevention and Treatment of DM” (Please see now the revised version of the manuscript)